# Insights into the Genetics and Signaling Pathways in Maturity-Onset Diabetes of the Young

**DOI:** 10.3390/ijms232112910

**Published:** 2022-10-26

**Authors:** Madalena Sousa, Teresa Rego, Jácome Bruges Armas

**Affiliations:** 1Serviço Especializado de Epidemiologia e Biologia Molecular (SEEBMO), Hospital de Santo Espírito da Ilha Terceira (HSEIT), 9700 Angra do Heroísmo, Portugal; 2CHRC—Comprehensive Health Research Centre, Faculdade de Ciências Médicas, Universidade Nova de Lisboa, 1169 Lisboa, Portugal; 3Serviço de Endocrinologia e Nutrição, Hospital de Santo Espírito da Ilha Terceira (HSEIT), 9700 Angra do Heroísmo, Portugal

**Keywords:** diabetes mellitus, genetics, monogenic diabetes, maturity-onset diabetes of the young

## Abstract

Diabetes Mellitus (DM) is a complex disease with a significant impact in today’s world. Studies have emphasized the crucial role of genetics in DM, unraveling the distinction of monogenic diabetes from the most common types that have been recognized over the years, such as type 1 diabetes (T1DM) and type 2 diabetes (T2DM). A literature search was carried out to scrutinize the subtypes of maturity-onset diabetes of the young (MODY), as well as the connection between the recognized genetic and molecular mechanisms responsible for such phenotypes. Thus far, 14 subtypes of MODY have been identified. Here, the authors review the pathophysiological and molecular pathways in which monogenic diabetes genes are involved. Despite being estimated to affect approximately 2% of all T2DM patients in Europe, the exact prevalence of MODY is still unknown, enhancing the need for research focused on biomarkers. Due to its impact in personalized medicine, a follow-up of associated complications, and genetic implications for siblings and offspring of affected individuals, it is imperative to diagnose the monogenic forms of DM accurately. Currently, advances in the genetics field has allowed for the recognition of new DM subtypes, which until now were considered to be slight variations of the typical forms. New molecular insights can define therapeutic strategies, aiming for the prevention, correction, or at least delay of β-cell dysfunction. Thus, it is imperative to act in the close interaction between genetics and clinical manifestations to improve diagnosis and individualize treatment.

## 1. Diabetes Mellitus—Background

In the last 40 years, an estimated 415 million adults were diagnosed with diabetes in 2015, as compared to 108 million in 1980, with both the incidence and prevalence of diabetes increasing globally [1]. Diabetes is a heterogeneous group of metabolic disorders, defined by persistent hyperglycemia due to both defects in insulin secretion and action, culminating in abnormal glucose metabolism with lifelong micro- and macro-vascular complications that develop from chronic hyperglycemia. It constitutes a significant cause of social, psychological, and financial burdens, along with an increased overall risk of premature death [1].

Over the past decades, increasing contributions from genetic studies resulted in the identification of variants associated with the susceptibility to and protection from diabetes. Those contributions drastically changed the understanding of diabetes, with a consequent acknowledgment of intermediary manifestations of the disease, namely maturity-onset diabetes of the young (MODY), latent autoimmune diabetes in adults (LADA), Neonatal Diabetes (NDM), Maternally Inherited Diabetes and Deafness (MIDD), and Gestational Diabetes Mellitus, in addition to the traditional and common subdivisions, namely type 1 diabetes (T1D) and type 2 diabetes (T2D).

This review will succinctly address the signaling pathways and respective genes affected in the various subtypes of monogenic diabetes, namely Maturity Onset Diabetes of the Young (MODY), as well as in an intermediary form of Diabetes, namely the Latent Autoimmune Diabetes in Adult (LADA).

## 2. Maturity Onset Diabetes of the Young (MODY)

Diabetes with early-onset hyperglycemia diagnosed in a patient under 25 years old, with an autosomal dominant transmission with at least three affected generations, a partially conserved pancreatic β-cell function, and the absence of autoantibodies are the characteristics of MODY [2]. This subtype of genetically transmitted diabetes is suspected to be the most frequent type of monogenic diabetes, with a prevalence of 21–45 in 1,000,000 children and 100 patients in 1,000,000 individuals.

Fourteen subtypes of MODY were identified and are currently acknowledged. Such MODY subtypes result from pathogenic variants in genes encoding proteins implicated in pancreatic β-cell normal function, enzymes involved in the insulin secretion pathway, and genes encoding proteins involved in gene expression regulation.

Approximately 70% of the MODY clinical cases in all the studied populations are caused by pathogenic variants in the genes GCK and HNF1A, MODY2 and MODY3, respectively.

Five percent of cases are due to pathogenic variants in the gene HNF4A, while the remaining 25% of cases originated from pathogenic variants in the genes PDX1/IPF1, HNF1B, NEUROD1, KLF11, CEL, PAX4, INS, BLK, KCNJ11, ABCC8, and APPL1 [3] (Table 1). Despite the various subtypes, the genetic cause remains undetermined in a considerable number of MODY individuals (MODYX), thus implying the existence of unknown genetic causes of diabetes in young patients. Depending on the gene affected with the pathogenic variant, different heterogeneous clinical features, such as the age of onset, response to treatment, and extra-pancreatic manifestations are observed.

The frequency of the genetic subtypes in populations is variable and dependent on the geographical origin of the individual and the regularity of screening for diabetes [4].

### Hepatocyte Nuclear Factors 1 and 4

The hepatocyte nuclear factors (HNFs) were first identified as liver-specific transcription factors, however, over the years, several findings showed its expression in other tissues and organs, including the developing embryo, pancreas, kidney, and intestines. HNFs are classified into four families, namely HNF1, FOXA (or HNF3), HNF4, and ONECUT ([OC] or HNF6), each characterized by distinct regions that correspond to functional domains.

The HNF1 family comprises HNF1A, and HNF1B genes, each encoding three isoforms, (A, B and C) with tissue-specific roles. HNF1A can be expressed as three alternatively processed transcripts, whose expression is likely to be regulated during development. In humans, HNF1A (A) is the major isoform in the adult liver, kidney, and fetal pancreas, whereas HNF1A (B) predominates in the adult pancreas.

HNF4 belongs to the orphan nuclear receptor family and comprises HNF4A and HNF4C. HNF4A is encoded by two developmentally regulated promoters (P1 and P2), and twelve isoforms (P1-derived HNF4A1-6 and P2-derived HNF4A7-12) can be created by the mechanisms of differential promoter usage and alternative splicing. The different isoforms are expressed in a temporal and tissue-specific manner, with the P1-driven HNF4A isoforms being liver-specific, whereas the P2-driven HNF4A isoforms are pancreas-specific (as reviewed in Reference [5]).

Increasing evidence has shown that several of the HNFs transcription factors not only control the cellular development and function by themselves, but are also co-dependent and part of a shared regulatory network, with tissue and development state-dependent roles.

During embryonic development, HNF1B was found to be a strong regulator of P1-driven HNF4A, together with GATA6, a fact also verified in liver development [22]. In the liver primordium forms of embryonic day (E) E8.5–9, besides inducing liver differentiation, HNF1B also cooperates with GATA6 to strongly stimulate the HNF4A promoter. As the liver differentiation proceeds, the expression of HNF1A increases and is accompanied by decreased levels of HNF1B until its inexistence in adulthood, which regulates the expression of the HNF4A gene expression in liver cells. However, studies show that HNF4A binds onto the proximal promoter of the HNF1A gene, via an evolutionarily conserved *cis* sequence element, regulating the HNF1A transcription in hepatocytes [66,67]. In vitro studies ascertained that point mutations in the HNF4A binding site of the HNF1A promoter reduce HNF4A binding, leading to reduced activation of the HNF1A gene expression [68].

Interestingly, HNF4A was found to bind to more than 40% of the promoters of actively transcribed genes in human hepatocytes, and occupies most of the promoters bound by HNF1A and OC, thus suggesting that HNF4A plays a central role in the transcriptional program of the liver.

Regarding the differentiation of liver cells into hepatocytes and biliary cells, OC1 activates HNF1B expression in the liver, starting the intrahepatic bile duct morphogenesis, which was proven by the diminished HNF1B expression and unsuccessful development of the gallbladder and severe extra and intrahepatic bile ducts in Oc1-null mice.

In the pancreas, HNF1A, OC1, and PDX1 are known to regulate the expression of HNF4a, specifically the HNF4a P2 promoter isoform, being the HNF4A7-9 isoforms that are active in pancreatic β cells.

In addition, HNF4A is also an essential activator of HNF1A gene expression by recruiting transcriptional coactivators. Conversely to what is observed in the liver differentiation, the Onecut family members Oc1 and Oc2, activated by HNF1B, are also involved in initiating the expression of the key pancreatic transcription factor PDX1, which is critical for the specification of pancreatic cell fate [69]. At later stages of development, Oc1 regulates pancreatic endocrine differentiation via Ngn3 expression [70].

Concerning the role of hepatocyte nuclear factors in the kidney, the most implicated genes are HNF1A and HNF1B, which are last that are in charge of controlling the expression of cystic genes, such as Pkd2 and Umod. HNF1B also controls the expression of Pkdh1 in tubular epithelial cells that are involved in tubulogenesis in the kidney, thus playing a role in nephron development.

While HNF-1β is not directly involved in primary cilium function, it regulates the expression of proteins that are localized at the cilium or that belong to biological processes mediated by cilia [71].

In turn, HNF1A is also expressed in the developing kidney, but is also a key regulator in maintaining glucose homeostasis through the regulation of the glucose-6-phosphatase system and the glucose transporter.

Despite the data already obtained from studies in mice, the physiological outcomes of HNF1A mutations in human kidneys require further investigation.

Mutations in HNF1A and HNF4A are the most frequently identified aetiologies of MODY, and combined, account for 62% of MODY cases, typically characterized by dysfunctional pancreatic β cells with accompanying liver and/or kidneys defects.

## 3. HNF1A-MODY (MODY3)

HNF1A-MODY accounts for approximately 30%–50% of the MODY cases, with nearly 414 mutations detected in 1247 families diagnosed. Such pathogenic variants are more frequently observed in exons 2 and 4, with a specific mutation (p.Gly292fs) accounting for nearly 10% to 15% of cases [72] Since patients with variants in the ending exons (8–10) are diagnosed, on average, eight years earlier than the ones with mutations in exons 1 to 6, studies show that the age at diagnosis can be partly pre-determined. The majority of pathogenic variants detected in the HNF1A gene are classified as mutations of high penetrance, once their presence is responsible for 63% of patients developing diabetes by the age of 25, 79% at 35 years old, and 96% by age 55 [73]. Heterozygous mutations in HNF1A are known to originate from MODY3, the most common form of MODY, characterized by a failure in glycaemic control with progressive impaired β-cell function. Individuals with heterozygous mutations are normoglycemic with the sufficient sensibility to insulin in the early stages, however, over time, typically occurring before the 25 years old, individuals with HNF1Apatohenic variants acquire an impaired glucose tolerance, and ultimately diabetes [5,22].

Similar to humans, HNF1A-null mice exhibit abnormal glucose-stimulated insulin secretion and develop diabetes two weeks after birth, expressing low levels of insulin and insulin-like growth-factor-1 (Igf1).

Therefore, the pathophysiology of HNF1A-MODY focuses on a severe reduction in insulin secretion in response to glucose. HNF1A has been shown to play a vital role as a transcription factor of the INS gene and GLUT2, encoded by the SLC2A2 gene, thus explaining the pancreatic-related disorders [14].

In the pancreatic β cell, GLUT2 acts as a glucose sensor that detects small changes in glucose levels leading to increased insulin secretion, and the lack of such transporters in the immature pancreas are likely to impact the β cell’s response to hyperglycemia. While the role of GLUT2 in mouse β cells has been well-established, the role of GLUT2 in human β cells has remained debatable. Unlike in rodent β cells, where GLUT2 is the predominantly expressed glucose transporter, human β cells predominantly express GLUT1 and GLUT3, thereby suggesting that it may not be the principal glucose transporter in human β cells. However, despite the purported irrelevance of GLUT2 in human β cells, the association of GLUT2 mutations in Fanconi–Bickel syndrome and diabetes pathology supports the imperative role played by GLUT2 in human β cells, regardless of its lower abundance [15].

Low et al. (2021) validated that GLUT2 deficiency, associated with reduced glucose uptake and ATP production, linked to a lack of insulin secretion observed in MODY3 patients, may possibly be caused by HNF1A mutations.

More specifically, the authors were not only able to disclose that HNF1A plays a role as a GLUT2 transcription factor, but also that the presence of a specific mutation in HNF1A, the H126D pathogenic variant, resulted in the failure of GLUT2 promoter activity upregulation, with decreased GLUT2 transcript and protein expression. Such observations show the modulation [14].

Aside from the characteristic progressive glycaemic control failure, patients with HNF1A mutations usually present two initial clinical manifestations: Reduced glucose reabsorption and glycosuria. Such phenotype can be elucidated by the effect of HNF1A loss-of-function mutations in extra-pancreatic tissues. The observed glycosuria—that is, excretion of glucose into the urine—was found to be associated with a low renal threshold for glucose as a result of deregulated expression of sodium–glucose cotransporter 2 (SGLT-2) and decreased glucose reabsorption in the proximal renal tubules. In the kidney, HNF1 α plays a vital role as a transcription factor in binding to the SGLT2 promoter and regulating its expression, as demonstrated by Pontoglio (2000) with HNF1A-null mice expressing reduced Sglt2 transcripts in the tubular cells [74]. The complete absence of HNF1A has been shown to result in several renal proximal tubular dysfunctions in the kidney, which can be linked to the clinical phenotype observed in human patients with Fanconi syndrome [74]. The poor glycemic control associated with MODY3 is responsible for amplified microvascular complications. Isomaa et al. examined the frequency of additional clinical traits, establishing that 47% of MODY3 patients displayed retinopathy, 19% presented nephropathy, and 4% were affected with neuropathy [75].

Additionally, more evidence confirmed the extra-pancreatic effect of HNF1A mutations. MODY3 patients suffer from renal dysplasia, growth hormone deficiency, and hypothyroidism, which is similar to homozygous HNF1A knockout mice, which exhibit stunted growth, reduced size, and weight 50%–60% less than their wild-type counterparts [76]. Less frequently, patients show an infantile uterus and unidentifiable ovaries, which are responsible for infertility [77].

## 4. HNF4A-MODY (MODY1)

HNF4A-MODY is present in 5%–10% of MODY patients, and is caused by more than 103 mutations identified in 173 families. Typically, pathogenic variants are often found in exons 7 and 8 of the HNF4A gene.

HNF4A-MODY patients usually exhibit a phenotype, clinical presentation, and sensitivity to sulfonylureas, which are similar to those found in HNF1A-MODY, therefore the genetic test to confirm the presence of a mutation in HNF4A is only performed if a pathogenic mutation in HNF1A is not identified. Inactivating mutations in HNF4A result in MODY1, in addition to the similar insulin secretory defects showed in MODY3, as explained by the fact that HNF1A regulates the expression of the HNF4A gene.

Studies show that HNF4a-null mice exhibit several liver-related alterations, such as affected hepatic epithelium, steatosis, severe disruption of gluconeogenesis, and hepatocellular carcinoma. Those alterations can be explained by the role that HNF4A plays as a transcription factor, regulating the expression of many genes involved in hepatic function by interaction with promoters of such genes, as apolipo- and metabolic proteins (APOA, APOB, PAH, and FABP1), as demonstrated by studies performed in liver-specific HNF4A-null mice. All the features mentioned pinpoint the crucial role played by HNF4A in the expression of genes implicated in regulating serum cholesterol levels in mice [78]. These observations are comparable to MODY1 patients that exhibit liver disorders, for example, reduced HDL cholesterol, apolipoprotein A1 and A2, and triglyceride levels, unlike LDL cholesterol levels, which are increased and constitute a differential phenotype between MODY1 and MODY3. It is worth mentioning that differential factors between HNF4A-MODY and HNF1A-MODY, besides altered cholesterol and triglycerides profiles, include the MODY1 patient’s progressive hyperglycemia associated with impaired insulin secretion that worsens with time and normal renal threshold for glucose. MODY1 patients typically exhibit an absence of glycosuria, increased birth weight (macrosomia), transient hypoglycemia and/or diazoxide-responsive hyperinsulinemia at birth, highlighting the different pathways that give rise to the MODY1 and MODY3 subtypes [9,79].

## 5. HNF1B-MODY (MODY5)

MODY5 accounts for 1%–5% of all cases and results from heterozygous HNF1B inactivating mutations, with more than 65 pathogenic variants being associated with MODY5 so far. Moreover, de novo mutations are frequent, comprising as much as half of all cases, meaning family history may be absent and approximately 28% of individuals present full allele deletion.

Due to HNF1B’s network and crucial role in embryo development, as well as liver, pancreas, and kidney differentiation, slight alterations in HNF1B expression results in multiple organ disorders. Starting in embryo development, HNF1B-null embryos fail to mature at the early stage of the blastocyst (E3.5) due to abnormal or absent extraembryonic endoderm, while HNF1B-null embryos rescued via tetraploid complementation failed to grow a ventral pancreas, with only a small dorsal pancreas [26]. As previously mentioned, the complex OC1-HNF1B cross-regulatory network in the pancreas development, where the expression of Oc1 in the pancreatic precursor cells is activated by HNF1B, leads to the expression of PDX1, which is critical for the specification of pancreatic cell fate, and at later stages, the regulation of pancreatic endocrine differentiation via the Ngn3 expression by Oc1. Therefore, mutations in HNF1B disturb the OC1-HNF1B, with consequent unsuccessful activation of such genes required for endocrine cell differentiation, with a consequent absence of endocrine cells and abnormal β-cell development [24].

Additionally, regarding the impact on the pancreas, impaired expression of HNF1B is also responsible for kidney alterations in relation to its target expression levels and function. As observed in MODY5 patients who are usually affected by renal cysts and diabetes (RCAD) syndrome, young mice with conditional knockout of HNF1B show polycystic kidneys, whereas knockout of HNF1B at P10 or later, results in significantly delayed cyst formation. The consequent results from the incapable binding of HNF1B to the proximal promoter of the mouse Pkhd1 gene contains an evolutionarily-conserved HNF-1-binding site located near a region of deoxyribonuclease hypersensitivity [25]. Regarding the Pkd2 gene, one of the cystic disease genes, responsible for the Ca^2+^-permeable cation channel Polycystin-2 (PC2), is strongly affected by HNF1B. PC2 interacts with polycystin-1 (PC1) in the primary cilium, and as the name suggests, facilitates Ca^2+^ entry, which is necessary for cAMP level regulation. Mutations in HNF-1β are associated with downregulation and consequently altered functioning of PC2, resulting in decreased Ca^2+^ entry, activation of the Ca^2+^-inhibitable adenylyl cyclases AC5 and AC6, and elevated cAMP levels in an indirect manner. Such increased levels of cAMP stimulate cell proliferation and fluid secretion, thus promoting cyst growth [23]. cAMP levels can also be increased directly by HNF1B regulation in phosphodiesterase 4C (PDE4C) expression. PDE4C, which catabolizes cAMP in the primary cilium, is downregulated in Hnf1b mutant kidney cells and mice [23].

Despite renal cysts being the most common abnormality, renal dysplasia, renal tract malformations, like horseshoe kidney, and/or familial hypoplasia glomerulocystic kidney disease have been reported, which in some severe cases ultimately led to end-stage renal failure. Moreover, less than 6% of HNF1B-MODY patients had a normal renal function and about half had end-stage renal failure.

Additionally, occasional genital tract abnormalities like vaginal aplasia or azoospermia, have also been reported, but penetrance is incomplete. Other associated anomalies are abnormal liver function, gallbladder dysfunction, hyperuricemia, and hypomagnesemia [27].

Regardless of the similarity between HNF1A and HNF1B, which share a highly conserved DNA-binding domain and a more divergent C-terminal transactivation domain, and could act as either homodimers or as heterodimers, the mechanisms by which mutations in HNF1B are responsible for the development of MODY5, phenotype and treatment are diverse, and are not entirely comparable.

Unlike the other MODY subtypes originated by alteration in HNF genes, in approximately 50% of HNF1B mutation carriers, diabetes results from a combination of β-cell dysfunction and insulin resistance.

Haumaitre et al. showed that the truncated R112fsdel and P472fsins, which causes a frameshift and a truncated protein-lacking part of the POU-specific domain (POUS), and premature stop codon and the insertion of 35 novel amino acids at the C-terminus of the transactivation domain, respectively, resulted in truncated protein by the formation of non-functional heterodimers and decreased transactivation capacity [80].

The aforementioned mechanisms can be responsible for early-onset diabetes in MODY5 patients, such as GLUT2 deficiency associated with reduced glucose uptake and diminished insulin secretion. Several studies have shown the connection between HNF1B mutations, namely R112fsdel or P472fsins (which disrupt its DNA binding domain), GLUT2, a potential direct target of HNF1B, and the MODY5 phenotype [80,81].

Moreover, MODY5 patients present a reduced insulin reserve, requiring early insulin therapy, unlike MODY1 and the three patients that respond to sulfonylureas treatment [69,80,82,83].

## 6. Glucokinase

GCK, also known as hexokinase IV or hexokinase D, is a product of the GCK gene and generates three tissue-specific isoforms via alternative splicing: Transcript variant 1 expressed in pancreatic β-cells, and transcript variants 2 and 3 expressed in the liver. The GCK enzyme plays an essential role in glucose homeostasis, acting as a highly sensitive glucose sensor, reaching its maximum activity with hyperglycemia. Due to its low affinity toward glucose and cooperative kinetics, it acts as a highly sensitive glucose sensor in cells and catalyzes the transfer of a phosphate from ATP to glucose to four generate glucose-6-phosphate (G6P). GCK activity peaks with hyperglycemia, being directly proportional to ambient glucose concentration [12,13]. In the liver, GCK synthesis is induced by insulin, thereby reflecting the organism’s nutritive state, with correspondingly high and very low GCK concentrations, contrary to glucagon, which suppresses GCK expression, while in β-cells, GCK is constitutively expressed, regardless of the body’s nutritive state, and by extension, insulin levels [11].

In β-cells, and similarly in hepatocytes, GCK-generated G6P undergoes glycolysis, yielding ATP. Such increasing levels potentiate numerous intracellular alterations with the closure of ATP-sensitive potassium channels and plasma membrane depolarization, thus facilitating insulin release [10,11]. Specifically in the liver, GCK regulatory protein (GKRP) binds and inhibits GCK in the nucleus, however increasing glucose concentration favors GCK release from GKRP, allowing GCK-glucose binding with glucose phosphorylation. Due to its role in both glucose metabolism and insulin secretion, it is no surprise that hyperglycemia and hypoglycemia constitute the more common phenotypes of GCK mutations [13].

## 7. GCK—MODY (MODY2)

To date, nearly 600 mutations have been associated with MODY2 in 1441 families and are most frequently detected in exons 7 and 9 [13].

Asymptomatic mildly stable hyperglycemia, present from birth, characterize GCK-MODY or MODY2. Such asymptomatic and non-progressive, hyperglycemia often remains undetected or misdiagnosed as T2DM or gestational diabetes (GDM) [3,12].

Mild and minor phenotypes result from heterozygous loss-of-function mutations, as stated by Grupe et al. based on mutant GCK heterozygotes mice, which develop mild early-onset diabetes that resembles GCK-MODY in humans [84].

Heterozygous pathogenic variants in the GCK gene ultimately leads to alterations in the GCK conformational state, causing a decreased phosphorylation rate, consequent impairment of glycogen synthesis. Additionally, such mutations are responsible for two differential effects, depending on the target cells. In hepatocytes, a blockage of postprandial glucose regulation occurs, while in β-cells, a diminished insulin secretion regulation with a new and higher glycaemic threshold for insulin release being established is observed.

An example of such a mechanism is the c.766G>A (p.Glu256Lys) variant [85,86]. This variant has been reported in numerous countries, and since Glu256 is located in GCK’s active site, conformational changes induced in GCK’s active site, as well as the whole structure, resulted in decreased glucose binding and a downstream loss of catalytic activity, thus explaining the hyperglycaemic phenotype [13].

Characterized by total GCK deficiency, MODY2 patients with compound heterozygous loss-of-function mutations display permanent insulin-requiring diabetes mellitus (PNDM) with neonatal onset. This more severe phenotype is also observed in individuals with homozygous mutations that cause GCK loss-of-function, although such mutations are witnessed in a small percentage of MODY2 individuals [13].

Hypoglycemia has also been established in MODY2 individuals and is associated with hyperinsulinemia when GCK changes from an inactive super-open conformation to a catalytically active closed conformation, even at lower glucose concentrations [12,13,87].

Unlike MODY1 and MODY3, treatment with oral hypoglycemic agents (OHA) or insulin therapy in MODY2 may be ineffective, since GCK mutations result in a deficient recognition of glucose. Therefore, the administration of exogenous insulin may trigger a compensatory response, with a decreased secretion of endogenous insulin [88].

The exception to insulin therapy in MODY2 is for pregnant women, in whom higher-than-standard doses may be required to prevent fetal overgrowth. In the case of pregnant women presenting heterozygous loss-of-function mutations and an unaffected child, the increased insulin secretion and insulin-stimulated growth secondary to maternal hyperglycemia can increase the risk of fetal macrosomia. However, if the baby inherits the mutation from the father and the mother is unaffected, due to the high glucose threshold, there will not be enough glucose to stimulate the appropriate insulin secretion for normal fetal growth, and the child will be born underweight. If both the mother and fetus carry mutations, the baby will have the necessary glucose to stimulate the proper insulin secretion for healthy fetal growth [89].

Unlike the elevated number of GCK-MODY cases is observed in European countries, explained by the increased number of routine monitoring in pediatric cases, pregnancy, and asymptomatic young individuals, the exact occurrence of GCK-MODY in different geographic locations and ethnic groups is poorly known [90,91,92,93,94,95,96,97,98]. Large-scale studies in different ethnic groups and more awareness to the mild symptoms, seem to be the path to recognize GCK-MODY.

## 8. Pancreatic Duodenal Homeobox 1

The β cell-enriched pancreatic duodenal homeobox 1 (PDX1) gene, also known as insulin promoter factor-1 (IPF1), is a transcription factor required for the embryonic development of the pancreas and has long been established by the observation of pancreatic agenesis in humans with homozygous mutations in the PDX1 gene. This is similar to what is observed in PDX1 null mice [99].

Multiple other aspects of mature β cell function are influenced by PDX1 expression, including insulin secretion and mitochondrial metabolism, by the transcriptional regulation of the endocrine pancreas-specific genes in adults, such as insulin (INS), glucose transporter-2 (GLUT2), and glucokinase (GCK) in β-cells, and somatostatin in δ-cells [17].

Pancreatic β-cell-specific PDX1 deficiency in mice leads to the reduced expression of insulin and SLC2A2, causing maturity-onset diabetes [16]. Alongside its importance in controlling metabolic mechanisms, PDX1 turns out to be a major player in preserving an adequate pool of healthy β-cells in adults, by adjusting the islet mass, architecture, and plasticity. Such regulation is only achieved by interfering in pathways that are shared with all cell types and are known as neogenesis, differentiation, and apoptosis, processes indirectly activated by the metabolic cell state by several well-known molecular pathways, such as Insulin/Igf signaling and glucose and fatty acid pathways. PDX1 is known to affect the insulin/Igf signaling cascade through the forkhead transcription factor Foxo1 and glycogen synthase kinase-3β (Gsk-3β). Insulin (or Igf1) inhibits the actions of Foxo1 by its phosphorylation, relieving the Foxo1 inhibition of PDX1 and inducing its movement from the nucleus to the cytoplasm.

Shortly after birth, the endocrine pancreas undergoes remodeling through a process that involves substantial apoptosis and β-cell replication [100]. The mechanisms responsible for β-cell apoptosis are complex.

Insulin/Igf signaling is an important regulator of β-cell growth and proliferation acting through the Irs1/Irs2-PI3K-Akt pathway. Over the years, the various players of this cascade were manipulated, allowing the scientific community to understand the impact of each impact in β-cells. Data indicate that loss of functional receptors for insulin in β-cell leads primarily to profound defects in postnatal β-cell growth in mice with a specific deletion of the β-cell insulin receptor (Ir) exhibiting a progressively impaired glucose tolerance, reduced β-cell mass, and islet number, while mice in which the Irs2 gene has been inactivated develop β-cell failure due to decreased proliferation and an increased rate of apoptosis [101,102].

Additionally, reduced levels of Foxo1 in diabetic Irs2−/− mice crossed with Foxo1+/− mice were able to moderately salvage the phenotype with a concomitant increase in PDX1 expression levels, suggesting that insulin and/or Igf regulate β-cell mass by relieving the Foxo1 inhibition of PDX1 expression [103]. Contrarily, Igf-induced β-cell proliferation was shown to be blocked in two models of insulin resistance, where the presence of the mutant Foxo1 transgene is retained in the nucleus and thus inhibits the expression of PDX1 [104].

The decrease in insulin/Igf signaling affects not only Foxo1, but also Gsk-3β function, and as evidenced by Tanabe et al., Gsk-3β controls β-cell mass in insulin-resistant diabetic models where the loss of one allele of Gsk-3β preserved β-cell mass and prevented diabetes in Irs2−/− mice. Diminished levels of Gsk-3β decreased the phosphorylation of Ser61 and Ser66 of PDX1 via Gsk-3β, rescuing PDX1 from proteosomal degradation, thereby increasing its half-life. In conclusion, the decrease in insulin/Igf signaling induces Foxo1 and Gsk-3β function, resulting in impaired replication and enhanced β-cell death at transcription and protein levels, respectively, ultimately leading to postnatal β-cell loss and diabetes [105].

The reason why PDX1 expression is important for β-cell maintenance has been a question for several years. Sachdeva et al. observed that PDX1 regulates a wide range of genes involved in diverse functions of the ER and PDX1-deficient β-cell showed evidence of endoplasmic reticulum (ER) stress with further β-cell susceptibility to ER stress-associated apoptosis. These findings suggest that PDX1 deficiency leads to a failure of beta cell compensation for insulin resistance, at least in part by impairing critical functions of the ER [106]. In addition to ER normal function, PDX1 has been shown to interfere in regular mitochondria function in pancreatic β-cells, with PDX1 being crucial in guiding the autophagosome-lysosome fusion during mitophagy [107]. Therefore, pathogenic variants in PDX1 may disturb mytochondiral function, contributing to β-cell failure, insulin resistance, and diabetes [108,109].

## 9. PDX1-MODY (MODY 4)

While pancreatic agenesis in human subjects is attributable to homozygosity for an inactivating mutation of the PDX1 gene, heterozygous carriers of PDX1 pathogenic variants develop MODY 4.

MODY4 was first described by Stoffers et al., who reported a homozygous single cytosine deletion within codon 63 (Pro63fsdelC) of the human PDX1 gene, previously attributed to the PNDM syndrome. Despite the similarity between MODY4 and PNDM pancreatic exocrine insufficiency, Stoffers and colleagues rectified this heterozygous pathogenic mutation as MODY4-causing. It is considered to be family-related when individuals developed diabetes over six generations, with an average age at onset of 35 years. Six of eight affected heterozygotes were treated with diet or oral hypoglycaemic agents and lacked ketosis or other indications of severe insulin deficiency.

Two pathogenic variants, E164D and E178K in the human PDX1 gene, also individually lead to PNDM and pancreatic exocrine insufficiency [110,111].

In 2011, in a family where the parents were carriers of the heterozygous form Pro63fsx60 of pathogenic variant, Fajans et al. found the presence of the homozygous form in a child with neonatal diabetes and exocrine pancreatic insufficiency. The authors established that in the carriers of this mutation, the onset of diabetes may occur at more advanced ages (around the age of 35), compared to other MODY subtypes [18]. Gragnoli et al. detected a Pro to Thr substitution (P33T) in the IPF1 transactivation domain, in an Italian family, with the clinical phenotype going from gestational diabetes, namely MODY4 to T2DM [112].

## 10. Neurogenic Differentiation 1 (NEUROD1)

Neurogenic differentiation 1 (NEUROD1), also called BETA2, is a class II basic helix-loop-helix (bHLH) transcription factor that dimerizes with E47, a class I bHLH transcription factor with ubiquitous expression, to form a heterodimer. This dimerization activates the transcription of genes that contain a specific DNA sequence known as a consensus E-box-binding site, such as the insulin promoter, the promoter of the sulfonylurea receptor 1 (SUR1), glucokinase (GCK), the glucose-6-phosphatase catalytic subunit-related protein, and PAX6 [28,29,30,31,32]. All these genes encode important molecules in maintaining normal glucose homeostasis. This transcription factor is expressed in pancreatic islet endocrine cells, the intestine, and a subset of neurons in the central and peripheral nervous system, and it plays a crucial role in the normal development and maintenance of these tissues.

NEUROD1 has been detected in the mouse embryo as early as E9.5 in glucagon-positive cells. It is mostly restricted to β-cells at birth and was found to affect β-cell dysfunction. NEUROD1–null mice exhibit enhanced apoptosis and consequently lacked mature islets, displayed NDM, and only survived the first days of life [32]. Overall, NEUROD1 is required for the development and maintenance of fully functional mature β-cells. More specifically, pancreatic β-cell-specific NEUROD1-deficient mice exhibited severe glucotoxicity with chronic hyperglycemia and display a glucose metabolic profile like immature β-cells with an increased expression of glycolytic genes, elevated levels of lactate dehydrogenase. Therefore, pathogenic variants in NEUROD1 encoding genes are responsible for another MODY subtype, designated as MODY6. The mechanisms responsible for such a phenotype are still scarce, however some hypotheses have arisen.

P300/CREB binding protein (CBP) coactivator potentially activates NEUROD1/BETA2, and although the exact mechanism involved in the p300/CBP-mediated transcription is unclear, it may result from the promotion of a transcriptionally active state to targeted genes through its intrinsic histone acetyltransferase activity [113,114]. Considering that p300/CBP also modulates the activity of a few key activators involved in regulating cellular proliferation and differentiation, including the myogenic bHLH factors, it may be important for transcriptional signaling during the development of specialized pancreatic and for enteroendocrine cells involved in differentiated gene product expression [113,114,115,116].

Supplementary to the development and maintenance of pancreatic islets, studies have evidenced that NEUROD1 plays a principal role in neuronal elements, as shown by Schwab et al., who generated double knock-out mice by crossing NEUROD1- null mice to a Nex-null genetic background mice. Such animals displayed severe neurological disorders, including ataxia, and collapsed frequently due to the pronounced reduction in the size of the cerebellum and complete loss of the dentate gyrus of the hippocampus [117].

## 11. NEUROD1-MODY (MODY6)

Other than permanent neonatal diabetes, NEUROD1-MODY patients usually display a range of neurological abnormalities, which include physical alterations such as cerebellar hypoplasia, as well as cognition disability relating to learning difficulties, and an alteration in two of the five senses, including sensorineural deafness and visual impairment [118].

Very few cases of a homozygous mutation have been reported, and this kind of condition usually leads to neonatal diabetes [119]. Recently, Rubio-Cabezas et al. reported two cases with homozygous frameshift NEUROD1 mutations (c.364dupG; p.Asp122Glyfs*12 and c.427_428del; p.Leu143Alafs*55) and both mutations introduced a frameshift to produce a prematurely truncated protein lacking the activation domain at the C terminus. These patients were diagnosed with permanent diabetes, and both exhibited a normal morphological pancreas and normal exocrine functioning. Moreover, patients had severe neurological abnormalities, including developmental delay, cerebellar hypoplasia, sensorineural deafness, and visual impairment [120]. Recently, a homozygous missense NEUROD1 mutation (c.449T>A; p.I150N) was reported with the same phenotype [121].

Approximately 20 families have been reported so far with heterozygous loss-of-function mutations in *NEUROD1.* The first pathogenic variant E110K was reported in an Icelandic MODY6 family, and following the missense variants, S159P, H241Q, and R103P in *NEUROD1* were identified worldwide [122].

Among the 20 families reported, there are 86 mutation carriers, of which 68 (79.1%) are glucose intolerant, nevertheless the several subjects remain glucose tolerant. Therefore, the overall phenotype of MODY6 is a broad clinical spectrum that ranges in patients with typical MODY features, to the incomplete penetrance of diabetes.

In Japanese patients, MODY6 individuals developed diabetes at less than 15 years of age, as well as diabetic ketoacidosis with a defect of early-phase insulin secretion without insulin dependence, unlike for Europeans. The mechanism by which such variations occurs remains to be elucidated, however a possible explanation relies on the induction of SHP gene expression by high glucose concentrations. The small heterodimer partner (SHP) also denominated the nuclear receptor subfamily 0 (NR0B2) and plays an important role in the development of β cell dysfunction induced by glucotoxicity. It has been reported that high glucose concentrations induce SHP gene expression with a consequent downregulation of INS expression and secretion by inhibiting p300-mediated pancreatic duodenal homeobox factor 1 (PDX1) and NEUROD1-dependent transcriptional activity from the insulin promoter [123]. In summary, it is likely that patients with genetically low insulin secretory capacity may be at an increased risk of development of diabetic ketoacidosis, and under sustained hyperglycemia [118].

As mentioned, NEUROD1 forms a heterodimer with the ubiquitous HLH protein E47 to transactivate INS expression by binding to a critical E-box motif on the promoter. Most of the reported mutations are present in the bHLH domain or the transactivation domain, thus causing disruption of DNA recognition of downstream target genes. Mechanistically, pathogenic variants in the NEUROD1 gene, located in this domain, abolish the E-box binding activity of NEUROD1 and significantly compromise INS transcription in pancreatic β-cells. Alternatively, the transactivation domain interacts with the cellular coactivator p300, possibly affecting the stimulation of target gene activation [33].

MODY6 individuals are equally treated with insulin therapy and oral glucose-lowering agents or diet, showing that the optimal therapeutic approach is yet to be disclosed, since the related literature suggesting OAD or insulin therapy is scarce.

Along with the endocrine phenotype, MODY6 may also be accompanied with neurological abnormalities, such as intellectual disability, although this is very rarely.

## 12. Kruppel Like Factor 11 (KLF11)

Kruppel Like Factor 11 (KLF11), or the TGF-inducible transcription factor, encodes a KLF/Sp1 family transcription factor comprised of three C2H2 zinc finger domains, which bind to target DNA, and three transcriptional repressor domains that interact with cofactors. KLF11 suppresses the expression of several genes through binding to GC-rich sequences in its promoter regions, like the INS gene in islet cells, as other MODY subtypes already mentioned, being a glucose-induced regulator of INS, as stated by Bonnefond et al., with KLF11-null mice exhibiting lower serum insulin levels than WT mice [91]. In addition, KLF11 has been shown to participate in the tissue-specific regulation of pancreatic acinar by repressing SMAD Family Member 7 (SMAD7) and Superoxide dismutase 2 (SOD2) and catalase 1 promoter, suggesting a role in the clearance of free radicals, thus making cells more susceptible to oxidative stress [36].

KLF11 is expressed in human tissues, including pancreatic islet cells that act as a negative regulator of exocrine cell growth, and its inhibition is often associated with pancreatic malignancy, as observed in MODY7 patients [37].

## 13. MODY-KLF11 (MODY7)

In 2005, Neve and colleagues reported KLF11 as a causative gene for MODY7 with the identification of two KLF11 variants, p.Ala347Ser and p.Thr220Met, in individuals diagnosed with early-onset T2DM, which were shown to significantly impair the transcriptional activity of KLF11.

Over the years, several studies have allowed for the identification of new pathogenic variants associated with MODY7, leading to the development of late-onset diabetes. Ushijima et al. identified a heterozygous KLF11 (p.His418Gln) variant in a family that was clinically diagnosed with early childhood-onset diabetes [93]. The combination of several studies allowed for an understanding of KLF11 function and diabetes outcome. Utilizing cells transfected with KLF11-WT and mutant plasmid with (c.1061G > T) mutation, KLF11-C354F-transfected cells, the authors concluded that this pathogenic variant impaired insulin promoter regulation activity and insulin expression and secretion in pancreatic beta cells, even upon stimulation with high glucose when compared to KLF11-WT cells. Authors predicted that such a decreased expression of INS could be a consequence of exposure on the surface of the protein, altering the protein activity, suggesting that the site is located in a larger transcriptional blocking domain, thereby affecting the transcriptional functions of INS [124,125].

A link between KLF11 to the Pdx-1 transcriptional regulation was also observed by Fernandez-Zapico et al. The authors observed that KLF11 activates Pdx-1 through the conserved GC1 and GC2 elements of Area II, which represents the principal control domain for islet cell expression, and that the existence of pathogenic variants Q62R, T220M, or A347S potentially had a significant impact on Pdx-1 expression levels comprised of islet β-cells function, since Pdx-1 critically influences the development and differentiation in islet cell mass and function during embryogenesis, in addition to regulating insulin and glucose sensitivity.

The same study showed that KLF11 is both a transcriptional repressor, as well as an activator, and such transcription activity can be mediated by p300. As mentioned in regards to other MODY subtypes, this coactivator has been shown to have a powerful coregulatory activity in 90% of MODY genes, hypothesizing that p300 recruitment is affected in MODY-causing variants [126].

KLF11 was proposed as a cause of MODY, with a candidate gene approach, in 2005 and a possible mechanism of action was suggested for the variants via the gain of function, which causes increased KLF11 repression activity [127].

However, recently, Laver et al. (2022) examined variant-level genetic evidence (co-segregation with diabetes and frequency in the population) for published putative pathogenic variants after concern has been raised about whether variants in KLF11, PAX4, and BLK1 cause MODY.

Given the high frequency of KLF11 variants in the population, poor cosegregation with diabetes in the families, and a lack of enrichment of rare variants in a MODY cohort, the authors conclude that such variants were not disease-causing [128].

## 14. CEL

The carboxyl ester lipase (CEL) gene encodes the digestive CEL enzyme involved in digesting milk and hydrolyzing dietary esters in the duodenum and is responsible for the hydration and absorption of cholesterol and liposoluble molecules. CEL enzyme is mainly expressed in pancreatic acinar tissues and lactating mammary glands. It is secreted by the exocrine pancreas and is rerouted within the intestinal lumen to participate in the hydrolysis of dietary lipids, for example, cholesterol ester digestion.

CEL contains 11 exons, where the last exon has a guanine/cytosine-rich variable number of tandem repeats (VNTR) that consists of nearly identical 33-base pair segments, each encoding 11 amino acids [129]. The number of repeated segments within the VNTR may vary between three and 23. In all the studied populations, the most common CEL allele contained 16 repeats.

## 15. CEL-MODY (MODY 8)

In 2006, Ræder, H. et al. reported single-base deletion (DEL) in the exon 11 of the CEL gene, comprised of the VNTR region, c.1686delT and c.1785delC. Such identification was only possible by focusing the study on patients with deficient exocrine pancreatic function. Functional studies demonstrated that, in spite of the similar in vitro catalytic activity, the enzyme resultant of the mutated gene was more instable and less secreted. The authors were able to conclude that the exocrine pancreatic dysfunction observed was due to the pathogenic variants detected in patients that fulfilled the MODY criteria [41].

The current literature supports the involvement of CEL exocrine and endocrine pancreatic dysfunction associated with another MODY subtype, however the pathophysiological mechanisms underlying this connection are yet to be fully understood.

After this first study, many have followed and claimed to identify new CEL pathogenic variants, nonetheless only one was strongly associated with MODY8 in an Italian individual [41,130].

Additionally, to the identification of the first CEL VNTR single-bp deletions, Ræder et al. characterized the MODY8 patients. The authors observed that, in addition to pancreatic phenotypes, such as pancreatic exocrine dysfunction in early childhood, diabetes, and pancreatic cysts, MODY8 individuals developed clinical malabsorption, deficient absorption of nutrients, as well as pancreatic fatty tissue accumulation [41].

Animal studies failed to dissect the disease mechanism of MODY8, however in vitro studies were able to indicate that the CEL VNTR single-bp deletions contain a different and shorter tail region, altered biochemical properties, and reduced O-glycosylation potential, concluding that CEL protein is misfolded. Altered CEL protein has a high propensity to form both intracellular and extracellular aggregates with cellular reuptake followed by lysosomal degradation, leading to impaired pancreatic cell line viability. The misfolded CEL protein has a major impact on endoplasmic reticulum (ER) stress, the stimulus of the unfolded protein response, and subsequent apoptosis [39,40].

More recently, El Jellas et al. (2022) uncovered the existence of two new CEL VNTR single-base pair deletions in the proximal part at the exon 11, in two different families, from Sweden and Czech Republic displaying the criteria for MODY [129].

Therefore, MODY8 is associated with pancreatic atrophy, fibrosis, and lipomatosis, together with exocrine insufficiency and later endocrine dysfunction and diabetes.

## 16. PAX4

The Paired Box 4 transcription factor PAX 4 is a transcription factor member of the PAX family, functioning as a regulator of fetal development, insulin, glucagon, somatostatin, IAPP, and ghrelin promoter repressor. Inversely, PAX 4 transactivates c-myc and Bcl-xL promoters, which promote islet proliferation and protect cells from stress-induced apoptosis, respectively [43].

PAX4 has been shown to be a critical regulator of islet cell lineages differentiation.

At the development stage, PAX4 is expressed in the ventral spinal cord and pancreatic bud at e9.5 in mice and is necessary to maintain the expression of PDX1 and Nkx6.1. It is also necessary for normal pancreatic β-cell development. Later, PAX4 restricts its expression in pancreatic β- and δ-cells, achieving its maximum at e13.5 to e15.5, and then diminishing its expression. Heterozygous pathogenic variants result in few mature β- and δ-cells, and numerous abnormally clustered α-cells. Thus, PAX4 absence is characterized by supporting α-cells at the expense of β-/δ-cell lineages [99]. The abnormalities observed in PAX4 null mice are similar to those observed in mice with a targeted disruption of the insulin promoter factor 1.

Contrariwise, a recent in vivo study demonstrated that over-expression of wild-type PAX4 in mouse islet β-cells protected against streptozotocin-induced hyperglycemia and cytokine-induced b-cell apoptosis [131].

In mature cells, PAX4 is necessary for cell regeneration as mutations affecting its expression impair β-cells proliferation [131].

PAX4-MODY is a relatively rare MODY subtype, and is more common in Asian populations, like the one identified by Plengvidhya et al., who recognized PAX4 mutations in two patients of Thai origin, who did not present mutations in the other known MODY genes [46].

Despite efforts being made, PAX4-MODY still remains one of the less understood subtypes, mainly due to its clinical characteristics that compromise its diagnosis.

## 17. PAX4-MODY (MODY9)

Knowledge of the molecular basis of PAX4 mutations causing diabetes remains incomplete. It is recognized that PAX4 predominantly represses the glucagon promoter activity in α-cells and weakly inhibits insulin promoter activity in β-cells by its transcriptional factor role [132].

To determine whether PAX4 mutations contributed to MODY, more specifically in the Thai population, Plengvidhya et al. examined PAX4 coding sequences in 46 MODY probands lacking mutations in other known MODY genes. The authors observed the first association of mutations in PAX4 to MODY diabetes by finding two possible pathogenic mutations of PAX4, R164W and IVS7–1G>A in patients with MODY, but not in nondiabetic controls and healthy subjects [46]. The altered protein R164W resulted in decreased PAX4 repression activity, while the guanine to adenine change at IVS7-1G>A intronic variant disrupted mRNA splicing and resulted in an in-frame deletion p.Gln250del (exon 8) with the repression of both insulin and glucagon’s promoter in α-cells. Complementary studies indicated that this pathogenic variant enhanced cell susceptibility to apoptosis upon cytokine or high glucose exposure. Conversely, the forced expression of wild-type PAX4 has been shown to protect against cytokine-induced β-cell death in isolated human islets [45,46,47].

Later, two new pathogenic variants were found by the same authors, namely PAX4 R192H and (C.374–412 del 39), as reported by Jo et al. and associated with MODY9 [48].

In vitro studies showed that the transcriptional repressor capacity on human insulin and glucagon promoters was reduced in cell lineages transfected with PAX4 R192H plasmid when compared to those of wild-type PAX4, suggesting that PAX4 R192H polymorphism generated a protein with a defect in transcriptional repressor activities on its target genes, leading to β-cell dysfunction associated with MODY and the early onset-age of T2D.

Regarding the 39-bp deletion in exon 3, this pathogenic variant caused exon 3 skipping and a truncated protein lacking part of the homeodomain and repressor domain in the carboxy terminus. This defective protein failed to repress the insulin and glucagon promoters [48,133].

Over the years, other missense mutations, including p.Arg31Leu14 and p.Arg52Cys,15 were found in an Indian and Malay patient, respectively, and both exhibited clinical hallmarks of monogenic diabetes.

PAX4 mutations can be located in the paired domain, the homeodomain, or between the paired domain and homeodomain, thus damaging its transcriptional repressor activity.

Persistent and severe β-cell dysfunction, flexible clinical features, and ketosis-prone diabetes (KPD) characterize PAX4-related MODY 9.

Mauvais-Jarvis and colleagues first reported that PAX4 homozygous R133W and heterozygous R37W mutations are associated with KPD [134]. Subsequent studies performed by Balasubramanyam et al. concluded that approximately 30% of the KPD patient subgroup had variants in HNF1A, PDX1, and PAX4 genes, suggesting that these variants may be the origin of β-cell dysfunction in a fraction of patients with A-β KPD [44].

On the whole, most MODY patients are submitted to sulfonylureas, metformin, or insulin treatment.

Despite the incomplete knowledge of the molecular basis of PAX4 mutations causing diabetes, new evidence has shed a light on possible treatments. Recent studies documented the efficacy of GLP-1 receptor agonists and dipeptidyl peptidase-IV inhibitors in MODY9 patients through their glucagon-inhibiting actions [135,136,137,138].

As previously mentioned, Laver et al. (2022) recently examined variant-level genetic evidence (co-segregation with diabetes and frequency in the population) for published putative pathogenic variants after concern has been raised about whether variants in KLF11, PAX4, and BLK1 cause MODY. While Plengvidhya et al. used the control subjects from the same population as the case subjects and we now know that p.R192H is common in East Asians. Additionally, despite the authors observing the impairment of the repressor activity of PAX4 on the insulin and glucagon promoters, the authors disclosed that the impairment was relatively modest, thus the reduction may be insufficient to result in a clinical phenotype. Until now, no large MODY pedigrees with cosegregation for a variant in PAX4 have been described since the initial report [46,128].

## 18. Insulin (INS)

The INS gene encodes for the pro-insulin precursor of insulin, which is an hormone secreted by the β-cells of the pancreas. Insulin acts by stimulating glucose storage into glycogen and decreasing glucose production, and GLUT4 translocation stimulates glucose transport.

Insulin acts on a specific cell membrane receptor belonging to the receptor tyrosine kinase superfamily, activating a complex intracellular cascade. The two main pathways of insulin signaling originating from the insulin receptor-IRS node are the phosphatidylinositol 3-kinase (PI3K, a lipid kinase)/AKT (also known as PKB or protein kinase B) pathway and the Raf/Ras/MEK/MAPK (mitogen-activated protein kinase, also known as ERK or extracellular signal-regulated kinase) pathway. The metabolic effects of insulin are due to the PI3K pathway and are connected exclusively through IRS, while the control of cell growth and differentiation, as well as the regulation of gene expression are derived from the MAPK pathway in cooperation with the PI3K pathway [50].

The stimulation of glucose transport in adipose tissue and skeletal and cardiac muscle prevents postprandial hyperglycemia, which is accomplished through the translocation of the GLUT4 from intracellular vesicles to the plasma membrane.

Insulin signaling in the liver and β-cell has emerged as the major determinant in preventing type 2 diabetes through the integrative role of molecules like IRS2 and FOXO, thus preventing β-cell dedifferentiation. It has been reported that a mutation in this gene causes a defect in the nuclear factor kappa-light-chain enhancer of the activated β-cells (NF-κB) transcription factor, leading to the reduced structural stability of insulin molecule associated with a very rare form of the MODY subtype, designated as MODY 10. INS-related MODY (MODY 10) is also linked with neonatal diabetes [50].

Moreover, variants of the INS gene are also strongly associated with common T1D and T2D.

## 19. INS-MODY (MODY10)

Heterozygous pathogenic variants in the INS gene result in MODY 10, which is a monogenic form of diabetes. Such genetic alterations result in a severe folding defect, which is an abnormal response to unfolded proteins, ß-cell apoptosis, and variable-onset diabetes mellitus. Dominant misfolding mutations in the INS gene are a frequent cause of isolated (PNDM). Therefore, it is only normal to verify decreased β-cell mass and gradual loss of insulin secretion in individuals displaying INS mutations [51,52].

Molven et al. reported the *INS* pathogenic variants, namely c.137G>A (R46Q), in a MODY10 family and c.163C>T (R55C) in a T1D family displaying ketoacidosis and insulin dependency. In vitro studies showed that the R46Q mutation disrupted a critical hydrogen bond formation, impairing the insulin molecule stability [54].

Soon after, Garin et al. and Carmody et al. identified *INS* mutations outside the exonic regions that are usually associated with diabetes. For example, mutations such as heterozygous c.188–31G>A (259) and homozygous c.187+241G>A were found to cause PNDM [139]. The intronic c.188–31G>A mutation ultimately results in an aberrant transcript producing misfolded proteins to induce ER stress and β-cell death [139,140]. Later, the c.188–31G>A mutation was also reported to cause MODY in one family [141].

After this, Dusatkova and colleagues unraveled a novel heterozygous single nucleotide deletion (c.233delA) leading to a frameshift mutation (Q78fs) in the *INS* gene in a MODY family. This mutation produces an aberrant proinsulin that lacks the native structures of the C-peptide and α-chain [142].

Complications with MODY10 have been reported in a few families, such as mild proliferative diabetic retinopathy, neuropathy, peripheral neuropathy, and polycystic ovarian syndrome.

At the time of diagnosis, diet or OADs may be used as a treatment for patients with MODY, but they eventually become insulin dependent [33,122].

## 20. B-Lymphocyte Kinase (BLK)

Expressed in pancreatic β-cells, B-lymphocyte kinase (BLK), which belongs to the SRC proto-oncogenes family, encodes a tyrosine receptor protein that stimulates β-cells to produce and secrete insulin and is essential for thymopoiesis in immature T cells. The upregulation of Pdx1, one of the key modulators of β-cell function, as well as the up-regulation of the transcription factor Nkx6.1, involved in the control of glucose-stimulated insulin secretion in pancreatic β-cells, act as a BLK mediator. β-cell function and proliferation enhancement is most likely BLK-induced by the increased protein levels of Pdx-1, which directly promotes the expression of Nkx6.1 [55,56,57].

## 21. BLK-MODY (MODY11)

Not all carriers of BLK pathogenic variants exhibit diabetes, and thus BLK-MODY has incomplete penetrance. The reason why a small proportion of the mutation carriers remain normoglycemic is unclear and is thought to result from environmental as well as genetic modifiers [143]. Borowiec and colleagues observed that the penetrance of a specific haplotype (three mutations occurred as a haplotype) was higher among carriers with a BMI greater than or equal to 28. Therefore, β-cell abnormalities caused by this haplotype might only come to light when in the presence of a diabetogenic environment conferred by increased body weight [58].

Recently, Laver et al. examined variant-level genetic evidence (co-segregation with diabetes and frequency in the population) for published putative pathogenic variants after concern has been raised about whether variants in *BLK* cause MODY [128]. The only BLK coding variant (p.A71T) reported to cause MODY was later found to be very common in normoglycemic individuals, showing that the variant is too common to cause MODY, raising doubt over the aetiological role of BLK. Additionally, since the initial report, no MODY pedigrees with consegregation of BLK pathogenic variants have been described [144].

Given the lack of evidence for coding variants in BLK as a cause of MODY, it is unlikely that noncoding variants would be pathogenic [58,128].

## 22. ATP Binding Cassette Subfamily C Member 8 (ABCC8) and Potassium Inwardly Rectifying Channel Subfamily J Member 11 (KCNJ11)

The assembly between four pore-forming inwardly rectifying potassium channel subunits (Kir6.2) and four regulatory sulfonylurea receptor 1 (SUR1) subunits comprise the ATP-sensitive potassium (K_ATP_) channel in the pancreatic β-cell membrane.

ATP Binding Cassette Subfamily C Member 8 (ABCC8) gene encodes the sulfonylurea receptor 1 (SUR 1) subunit of ATP-sensitive potassium (K_ATP_) channel, and the Potassium Inwardly Rectifying Channel Subfamily J Member 11 (KCNJ11) gene encodes the Kir6.20 subunits.

The K_ATP_ channel is involved in the electrical activity of the plasma membrane, thereby regulating insulin secretion by coupling β-cell metabolism to calcium entry. Glucose levels in pancreatic β-cells ascend in the state of hyperglycemia, metabolized by the β-cell via glycolytic and mitochondrial metabolism, leading to a fall in MgADP levels and an elevated ATP production. Consequently, the K_ATP_ channel closes. As a consequence of the reduced K^+^ efflux, the membrane depolarizes, thus opening the voltage-gated Ca^2+^ channel, triggering electrical activity, Ca^2+^ influx, and insulin secretion [59]. Both activating and inactivating pathogenic variants of ABCC8 and KCNJ11 are associated with a variety of phenotypes, thus resulting in overactivity or underactivity of the K_ATP_ channel, with subsequent abnormal glucose metabolism. Inactivating mutations in KCNJ11 or ABCC8 result in congenital hyperinsulinism (CHI) by enhancing ATP binding to the channel, leading to K_ATP_ channel closure, membrane depolarization, and insulin overproduction in the β-cell. Regarding K_ATP_ channel-activating mutations, they either impair ATP binding to the channel Kir6.2 mutant and is impaired or enhanced by the binding of Mg-nucleotide to SUR1 mutant leading to K_ATP_ channel opening, membrane hyperpolarization, impaired insulin release, and can lead to NDM [59].

ABCC8 and KCNJ11 gene mutations are also responsible for MODY12 and MODY13, respectively, leading to impaired fasting glucose or impaired glucose tolerance, due to the disrupted subunit interaction, and to a history of neonatal diabetes.

## 23. ABCC8-MODY (MODY12)

The *ABCC8* gene is responsible for at least 1% of MODY cases in the literature and there are about 700 pathogenic variants of the *ABCC8* gene in the HGMD database, with more than half of them being missense and nonsense variations.

Bowman et al. first reported that MODY12 is caused by ABCC8 gene mutation in 2012. Until now, only 55 ABCC8 variants were associated with MODY12 [145].

Data showed that four patients were heterozygous for previously reported mutations and four novel mutations, E100K, G214R, Q485R, and N1245D, were identified. Only four probands fulfilled the MODY criteria, with two diagnosed after 25 years and one patient, who had no family history of diabetes as a result of a proven de novo mutation. The four unique mutations were found in susceptible MODY patients with diverse clinical manifestations associated with overweight or obesity and with no significant hypertriglyceridemia and hypercholesterolemia. Despite the residues being highly conserved, suggesting a pathogenic impact, the authors stated the need for functional studies to show that the mutations increase KATP channel activity and cause diabetes. Both activating and inactivating pathogenic variants of the ABCC8 gene were found to trigger MODY 12 [61].

Pathogenic variants in the *ABCC8* gene ultimately lead to membrane hyperpolarization and impaired insulin secretion, due to augmented binding of Mg-nucleotide to nucleotide-binding domains of SUR [60].

ABCC8 gene mutations can result in congenital hyperinsulinism, which can be caused by dominantly inherited inactivating mutations. As a result of activating mutations or recessive loss-of-function mutations, ABCC8 gene mutations can lead to other forms of monogenic diabetes, such as permanent or transient neonatal diabetes (PNDM or TNDM, respectively) [146].

Rafiq et al. proposed that, in adulthood, all ABCC8 mutation carriers could be switched to sulfonylureas since sulfonylureas specifically bind to the SUR1 subunit of the K_ATP_ channel and shut down the channel to release insulin in a non-ATP-dependent manner, with MODY being sensitive to sulfonylureas. In fact, the treatment switch from insulin to sulfonylureas was proven to improve patients’ glycemic control, as well as decrease the risk of hypoglycemia episodes [147].

## 24. KCNJ11-MODY (MODY13)

Molecularly, MODY 13 corresponds to the existence of activating mutations in KCNJ11, which is associated with decreased ATP sensitivity to the Kir6.2 subunit characterized by the prolonged open state of the channel and indirectly influencing ATP sensitivity, thereby compromising the insulin secretory response.

*KCNJ11* gene screening is currently indicated by guidelines in all patients who present with diabetes diagnosed before 6–12 months of age since some studies reported that families of patients with a transient or permanent form of NDM can also include individuals with childhood or later-onset diabetes. However, no previous study has described a family with a well-defined MODY due to a *KCNJ11* mutation.

Bonnefond et al., by focusing on variants of interest, found 69 mutations in KCNJ11 in the three affected relatives and not present in the control population. Subsequently, only one mutation (p.Glu227Lys in KCNJ11) co-segregated with diabetes in the family. Data confirmed that *KCNJ11* mutations can be associated with a large spectrum of diabetes phenotypes and cannot completely penetrant as one of the identified members of the French MODY family that carries the *KCNJ11* p.Glu227Lys mutation, has normal fasting plasma glucose level at 39 years. This large phenotype spectrum has also been reported in carriers of pathogenic variants in *ABCC8* and the *INS*, which together with *KCNJ11*, represent the most frequently mutated genes in patients with NDM. Other modifier, genetic effects such as epigenetics could explain the substantial difference in both diabetes onset and clinical expression between NDM and MODY patients [62].

Gloyn and colleagues revealed, in 2004, the existence of six new heterozygous missense mutations in 10 out of 29 patients, and among these four patients, exhibit p.Arg201His pathogenic variants. They concluded that neonatal diabetes was caused by heterozygous mutations of the KCNJ11, with variable onset and the severity of diabetes [148]. Later, another group of studies uncovered the occurrence of five different heterozygous mutations, including two novel mutations in the KCNJ11 gene in eight Italian patients, concluding that KCNJ11 gene mutations are the common cause of PNDM [149].

As with INS-related MODY (MODY 10) and ABCC8-related MODY (MODY 12), MODY 13 is associated with neonatal diabetes, which is also sensitive to sulfonylurea therapy.

## 25. Adaptor Protein, Phosphotyrosine Interacting with PH Domain and Leucine Zipper 1 (APPL1)

Adaptor protein, a phosphotyrosine interacting with PH domain and leucine zipper 1 (APPL1) gene, is widely expressed in all insulin target tissues and organs, including the liver, adipose tissue, and skeletal muscle, as well as the pancreas, which is also a regulator of cell proliferation. In pancreatic islets, APPL1 is abundantly expressed and acts as a physiological regulator of insulin secretion, binding to AKT2 (RAC-beta serine/threonine-protein kinase or AKT serine/threonine kinase 2), a key molecule in the insulin signaling pathway, thereby enhancing insulin-induced AKT2 activation and downstream signaling in the liver, skeletal muscle, adipocytes, and endothelium [64]. APPL1 plays different roles depending on the target cells, since, in skeletal muscle and adipocytes, APPL1 controls the translocation of GLUT4 to the plasma membrane, and in the liver activates AKT, inhibiting hepatic gluconeogenesis [64]. Regarding adipocytes, APPL1 executes the role of adaptor in the adiponectin signaling pathway, which is essential for anti-inflammatory and antidiabetic effects, thus enhancing insulin sensitivity. Decreased adiponectin levels are associated with obesity and metabolic syndrome.

In a mouse model of T1DM, investigators have demonstrated a negative action of APPL1 on the regulation of inflammation and apoptosis in β-cells of the pancreas [150].

Mutations in the APPL1 gene can cause protein loss-of-function, leading to MODY 14 phenotype.

## 26. APPL1-MODY (MODY 14)

MODY 14 is a rare subtype. Heterozygous loss-of-function mutations in this gene result in diminished insulin secretion in response to glucose stimulation and increasing β-cell apoptosis [151]. In 2015, Prudente et al. reported two loss-of-function mutations (c.1655T>A [p.Leu552∗] and c.280G>A [p.Asp94Asn]) in APPL1, identified by whole-exome sequencing in two large families with a high prevalence of diabetes. Both mutations caused APPL1 to lose function. The authors observed that the p.Leu552∗ pathogenic variant caused deletion of most of the PTB domain, thereby making APPL1 unable to bind to AKT and abolishing APPL1 protein expression in HepG2 transfected cells [65]. The missense p.Asp94Asn alteration affected the aspartic acid residue at position 94, located on the concave surface of the APPL1 BAR domain, and is highly conserved among various species, causing a noteworthy reduction of the insulin-stimulated AKT2 and GSK3β phosphorylation, in comparison to wild-type APPL1 transfection. In APPL1 WT cells, APPL1 binds to AKT2, which is a key molecule in the insulin signaling pathway, thereby enhancing insulin-induced AKT2 activation and downstream signaling and leading to insulin action and secretion. Therefore, these findings reaffirm the critical role of APPL1 in glucose homeostasis [65].

More recently, Ivanoshchuk and colleagues observed that rs11544593 may contribute to the earlier onset of carbohydrate metabolism disorders with an association of rs11544593 with blood glucose concentration revealed in the MODY group [152]. Schenck et al. detected that, in pathogenic variants in APPL1 in tissues where this gene is highly expressed, there is increased apoptosis [151].

## 27. Conclusions

Currently, monogenic diabetes diversity is more than merely recognized but is consistently pinpointed by reports that see it as a major player in both the disease’s risk onset and progression. MODY is a rare type of diabetes that is still difficult to identify and is largely underdiagnosed, despite the clear health implications for the individuals and their families. Effort has been made to clarify MODY’s underlying molecular mechanisms, and consequently to obtain information on personalized medicine with an accurate diagnosis, allowing individualized management and family screening.

## Figures and Tables

**Table 1 ijms-23-12910-t001:** Fourteen subtypes of MODY were identified and are currently acknowledged with their related information.

Subtype	Gene	Inheritance	Prevalence	Molecular Mechanisms	Pathophysiology	Clinical Information
MODY1	*HNF4A*	AD	Rare (5%)	Regulation of gene expression in the early liver; Regulation of HNF1A transcription in hepatocytes [5]; Regulation of apolipo and metabolic genes (APOA, APOB, PAH, FABP1) [6,7]	Abnormal embryo development (liver and pancreas dysfunction) [5]; Diminished insulin secretion capacity [8,9];	Transient neonatal hyperinsulinemia and hypoglycemia with associated macrosomia; Gestational diabetes; Progressive insulin secretory defect accompanied with tendency for microvascular complications; Low serum levels of triglycerides, Apo AI, AII and CIII; Marked sensitivity to sulfonylureas [8,9];
MODY2	*GCK*	AD	Common (30–50%)	Glucose phosphorylation [10,11,12,13]	Decreased glucose phosphorylation capacity; Decreased glycogen storage Diminished insulin secretion; β-cell dysfunction [13]	Stable, mild fasting hyperglycemia throughout life, with increased likelihood of glucose <55 mg/dL on oral glucose tolerance test; Typically asymptomatic;, with the diagnosis often incidental; Mild diabetes that generally does not require anti-diabetes medication. Managed with diet and exercise; Gestational diabetes; Variation in birth weight reviewed in [3];
MODY3	*HNF1A*	AD	Common (30–50%)	Regulation of gene expression in the early liver, kidney, intestine, and pancreas [5]; Regulation of INS and SLC2A2 gene expression [14]	GLUT2 defiency associated with reduced glucose uptake and ATP production; Diminished insulin secretion capacity [14,15]	Absence of antibodies against Langerhans’ islets; High penetrance; Transient neonatal, hyperinsulinemia and, hypoglycemia for some; Diminished renal threshold for glycosuria, accompanied with glycosuria; Progressive insulin secretory defect with microvascular abnormalities; Marked sensitivity to sulfonylureas reviewed in [3];
MODY4	*PDX1/* *IPF1*	AD	Very rare, <<1%	Regulation of INS, GLUT2, GCK, HNF4A and SLC2A2 gene expression [5,16,17]	Dysfunction of β cell maturation; Altered pancreatic development [18] Decreased insulin secretion capacity [19];	Diagnosis at 35 years old; requires oral anti-diabetes and insulin treatment; 1-bp deletion found in a child with agenesis of the pancreas belonging to an inbreeding family. Additionally, heterozygous relatives for this same disorder displayed an early-onset T2DM phenotype [18,19,20,21]
MODY5	*HNF1B*	AD	Rare	Cooperation with GATA6 for HNF4A expression [22]; Activation of OC1 and OC2 genes; Regulation of Pkdh1, Pkd2, Umod, and PC2 genes expression [23,24,25];	Abnormal embryo development Absence of endocrine cells; Abnormal β-cell development [24,26,27]	Renal pathologies and microvascular complications. No sensitivity to sulfonylureas and insulin therapy requirement reviewed in [3];
MODY6	*NEUROD1*	AD	Very rare, <<1%	Regulation of INS, SUR1, GCK, IGRP and PAX6 gene expression [28,29,30,31,32]	Pancreatic islet endocrine cells and enteroendocrine cells abnormalities [33].	Rare heterozygous mutations cause MODY among other pathologies such as PND and syndromic NDM [34,35];
MODY7	*KLF11*	AD	Very rare, <<1%	Regulation of INS, SMAD7, SOD2 and CAT1 gene expression [36]	Exocrine cell growth dysfunction; Pancreatic malignancy [37]	Similar phenotype to T2DM [38];
MODY8	*CEL*	Deletion of VNTR	Very rare, <<1%	Altered CEL protein intracellular and extracellular aggregates [39,40]	Decreased pancreatic endocrine and exocrine functions; Decreased viability of pancreatic cells [39,40]	Pathogenic variants in 2 Norwegian families associated with childhood-onset exocrine pancreatic dysfunction [41,42];
MODY9	*PAX4*	AD	Very rare, <<1%	Regulation of INS, IAPP, and ghrelin promoter’s repressor gene expression Regulation of Glucagon and somatostatin [43]	Decreased maturation and proliferation of β-cells [44,45,46,47]	Possible ketoacidosis; Three pathogenic variants detected in two Thai families and one Japanese family, both primarily diagnosed with early-onset T1DM and T2DM [48,49];
MODY10	*INS*	AD	Rare, <1%	Regulation of PI3K and ERK cascades [50]; Regulation of NF-ƘB transcription factor; Misfolding of proinsulin molecules [51,52]	Reduced structural stability of insulin molecule [50]; β-cell dysfunction; Decreased insulin secretion capacity [51,52]	Onset before age 20, insulin therapy and sulfonylureas usually required; Two pathogenic viants reported as causing INS-MODY [53,54];
MODY11	*BLK*	AD	Very rare, <<1%	Regulation of Nkx6.1 and PDX1 Promotion of insulin synthesis and secretion [55,56,57]	Reduction of insulin content; Desensitization of β-cells to glucose; Insulin secretion deficit [55,56,57]	Increased penetrance with increasing body mass index; Five mutations associated with MODY in 3 families of different ethnicities [58];
MODY12	*ABCC8*	AD	Very rare, <<1%	Ca^2+^ influx and β-cells metabolism [59]	Dysfunction of ATP-sensitive potassium channels; Decreased insulin secretion; [59,60]	Heterogeneous phenotype; Pathogenic variants cause a phenotype similar to MODY3 and MODY1 in patients with no pathogenic variants in *HNF1A* and *HNF4A* [61];
MODY13	*KCNJ11*	AD	Very rare, <<1%	Ca^2+^ influx and β-cells metabolism [59]	Dysfunction of ATP-sensitive potassium channels; Decreased insulin secretion; [59,60]	Heterogeneous phenotype; Two variants identified as causative of KCNJ11-MODY in 12 individuals from a French family, with age of diagnosis ranging from 13 to 59 years of age, a heterozygous alteration in a Japanese family with 4 affected individuals [62,63].
MODY14	*APPL1*	AD	Very rare, <<1%	Regulation of insulin secretion by AKT2 binding [64]	Loss of function and significant reduction of phosphorylation of AKT2 and GSK3β Diminished insulin secretion Increased of β-cells apoptosis [65]	Phenotype similar to T2D; Two heterozygous pathogenic variants in *APPL1* in 2 out of 60 families [65];

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
