# Peer review of "Insights into the Genetics and Signaling Pathways in Maturity-Onset Diabetes of the Young"

_ijms, 2022, doi:10.3390/ijms232112910_

Round 1
Reviewer 1 Report
In the manuscript, ijms-1917012 by Sousa et al, authors have reviewed the literature for scrutinizing the subtypes of maturity-onset diabetes of the young (MODY), as well as the connection between the associated genetic and molecular mechanisms. Authors have discussed the pathophysiological and molecular pathways associated with monogenic diabetes genes. This review is comprehensive and includes over 150 references. In general, the manuscript is well written.
There are the following points:
1. The abstract can be revised for understanding. It does say what the authors have reviewed, but somehow the purpose of this review article is not conveyed in the abstract.
2. Such a comprehensive review can be supplemented by a couple of schematics.
3. Heading and subheadings are not distinguishable.
4. Many headings have several small paragraphs ~ 2-3 lines. Is there any reason for that? If not, please consider consolidating them.
5. Mainly, MODY is discussed comprehensively. Accordingly, is Line 50 accurate? Please check.
6. Please consider putting the full form in the first instance. For example, HNF is not defined for the first time.
7. Line 22. “An effort has also been made to unravel MODY’s underlying molecular mechanisms.” Not clear. Is this sentence referring to this review discussion as an effort? A similar thing in line 1008 as well.
8. The conclusion section can have some notes about future perspectives too.
Author Response
Dear Reviewer,
We appreciate your comments and addressed them in our final version.
Best Regards,
Madalena Sousa
Reviewer 2 Report
I found this paper is a valuable review of MODY, which is difficult to diagnose and often underdiagnosed. This paper is a valuable resource for accurate diagnosis and personalized medicine in the future. I would like to express one point of concern.
Comment #1 The subheadings are mixed with gene names and MODY types, which seems a bit confusing. How about you summarize them under a subheading for each type of MODY.
Author Response
Dear Reviewer,
We appreciate your comments and addressed them in our final version.
Regarding the “Comment #1 The subheadings are mixed with gene names and MODY types, which seems a bit confusing. How about you summarize them under a subheading for each type of MODY.”, we did our best to divide each section, subtitling the title in the subsection regarding the specific MODY subtype. We hope that it improves the understanding of the manuscript.